# Viral GPCR US28 can signal in response to chemokine agonists of nearly unlimited structural degeneracy

**Timothy F Miles[1,2†], Katja Spiess[3†], Kevin M Jude[1,2†], Naotaka Tsutsumi[1,2†], John S Burg[1,2†], Jessica R Ingram[4], Deepa Waghray[1,2], Gertrud M Hjorto[3], Olav Larsen[3], Hidde L Ploegh[5], Mette M Rosenkilde[3], K Christopher Garcia[1,2,6]\***

[1]Department of Molecular and Cellular Physiology, Stanford University School of Medicine, Stanford, United States; [2]Department of Structural Biology, Stanford University School of Medicine, Stanford, United States; [3]Laboratory for Molecular Pharmacology, Department of Biomedical Sciences, Faculty of Health and Medical Science, University of Copenhagen, Denmark, Europe; [4]Department of Cancer Immunology and Virology, Dana Farber Cancer Institute, Boston, United States; [5]Program in Cellular and Molecular Medicine, Boston Children's Hospital, Boston, United States; [6]Howard Hughes Medical Institute, Stanford University School of Medicine, Stanford, United States

**\*For correspondence:**
kcgarcia@stanford.edu

[†]These authors contributed equally to this work

**Competing interests:** The authors declare that no competing interests exist.

**Abstract** Human cytomegalovirus has hijacked and evolved a human G-protein-coupled receptor into US28, which functions as a promiscuous chemokine 'sink' to facilitate evasion of host immune responses. To probe the molecular basis of US28's unique ligand cross-reactivity, we deep-sequenced CX3CL1 chemokine libraries selected on 'molecular casts' of the US28 active-state and find that US28 can engage thousands of distinct chemokine sequences, many of which elicit diverse signaling outcomes. The structure of a G-protein-biased CX3CL1-variant in complex with US28 revealed an entirely unique chemokine amino terminal peptide conformation and remodeled constellation of receptor-ligand interactions. Receptor signaling, however, is remarkably robust to mutational disruption of these interactions. Thus, US28 accommodates and functionally discriminates amongst highly degenerate chemokine sequences by sensing the steric bulk of the ligands, which distort both receptor extracellular loops and the walls of the ligand binding pocket to varying degrees, rather than requiring sequence-specific bonding chemistries for recognition and signaling.
DOI: https://doi.org/10.7554/eLife.35850.001

## Introduction

Chemokines are small immunomodulatory proteins that act through a large family of G-protein-coupled receptors (GPCR) (*Charo and Ransohoff, 2006*; *Proudfoot, 2002*). More than 40 chemokines and over 20 chemokine receptors are encoded in the human genome, and there is extensive receptor-ligand cross-reactivity, which can manifest as preferential signaling via either G protein or β-arrestin, a process called biased agonism (*Steen et al., 2014*). Human cytomegalovirus (CMV) has 'hijacked' a relatively ligand-specific human GPCR, and repurposed it through evolution to serve as a highly cross-reactive 'chemokine sink' as a mechanism to subvert host immunity (*Randolph-Habecker et al., 2002*). US28 binds with high affinity to many CC-type chemokines, in addition to CX3CL1 (*Kledal et al., 1998*). The molecular mechanisms for how US28 can engage and respond to such a wide range of chemokines are not understood; indeed, most receptor-ligand interactions are

characterized by a high degree of specificity. Furthermore, it remains unclear to what extent, and how, US28 has the capacity to signal differentially in response to these chemokines.

A crystal structure of CX3CL1 bound to US28 (*Burg et al., 2015*) showed that the chemokine bound through a two-site interaction mechanism (*Allen et al., 2007*; *Monteclaro and Charo, 1996*; *Thiele and Rosenkilde, 2014*) that is generally shared by other chemokine GPCRs (*Wu et al., 2010*; *Wescott et al., 2016*; *Zheng et al., 2017*). At Site 1, the receptor N-terminal region binds a groove on the globular body of the chemokine. At Site 2, the chemokine N-terminal peptide binds within a deep pocket formed by the receptor transmembrane helices (TMs) that is believed to function as the receptor activation switch. The structure also revealed extensive contacts with receptor extracellular loops (ECLs), coined Site 1.5, of unknown function. Given that US28 has a unique capacity to bind many diverse chemokines with high affinity (*Kledal et al., 1998*), we sought to determine: 1- the breadth of ligand promiscuity that can be accommodated, 2- how sequence differences impact signaling, and 3- the structural properties of this interaction that enable such cross-reactivity.

## Results and discussion

### Chemokine-induced US28 signaling

Radioligand studies reveal a complex network of noncompetitive binding by these chemokines (*Figure 1a*). Whereas CX3CL1 (Fractalkine) and vMIP-II (a broad-spectrum CC-chemokine antagonist encoded by Kaposi's sarcoma-associated herpes virus (KSHV) [*Kledal et al., 1997*]), bind US28 competitively, CCL3 (MIP1α) and CCL5 (RANTES) are only competitive with themselves. This manner of 'orthosteric allostery' is well established among chemokine receptor antagonists and suggests significant differences in the nature of receptor engagement, with the potential for differences in chemokine modulation of US28 activity (*Kufareva et al., 2017*). Although isolated studies have attempted to delineate the signaling effects of individual chemokines at US28, no comprehensive effort has been made to compare chemokine-induced activity among US28's many ligands or to determine whether they act as biased ligands. US28 signals constitutively via the G protein Gq (*Casarosa et al., 2001*), and cells expressing US28 exhibit constitutive cell migration (*Streblow et al., 1999*), which is thought to principally occur through β-arrestin signaling (*DeFea, 2007*). We sought to establish the capacity of natural and chimeric chemokine ligands of US28 to modulate this constitutive activity. We tested CX3CL1, CCL5, CCL3, and vMIP-II, as well as N-terminal chimeric chemokines, for their effects on Gq-mediated calcium flux and cell migration, which serves as a proxy for β-arrestin signaling.

These chemokines (four natural and two chimeric) produce a broad range of ligand-induced signaling activities, on top of the receptors' constitutive activity. The high levels of constitutive signaling and β-arrestin recruitment (*Figure 1b and d*) complicate precise quantitation and leave a narrow dynamic range in which to directly observe ligand effects, but nevertheless we were able to detect significant changes in US28 activity upon addition of chemokine. Comparing Gq-mediated calcium flux and migration responses, we find that US28 is capable of highly plastic but ligand-specific signaling activities that qualitatively suggest capacity for biased signaling (*Figure 1c and e* and *Figure 1— figure supplements 1* and *2*). vMIP-II binds US28 with high affinity while having no effect on the receptor's activity, acting as a neutral antagonist (*Figure 1c and e* and *Figure 1—figure supplement 3*). CCL3 increases $Ca^{2+}$ signaling while migration remains unaffected, thus appearing to be a G-protein-biased agonist. CCL5 and CX3CL1 potentiate both $Ca^{2+}$ signaling and cell migration, with CX3CL1 exhibiting more prominent cell migration. Despite prior reports of Gq inverse agonism by CX3CL1 (*Tschammer, 2014*; *Waldhoer et al., 2003*), we observe US28-mediated CX3CL1 inverse agonism only in IP3 assays with COS-7 cells (*Figure 1b*). Chimeric chemokines in which the globular body of CX3CL1 was appended to the N-terminus of vMIP-II (NVF) or CCL5 (N5F) displayed weak calcium flux and moderate cell migration, diminishing the effects of CCL5 while converting vMIP-II from an antagonist to an agonist (*Figure 1c and e*). It is unclear whether this signaling behavior arises from direct contacts by the chemokine globular body or indirect contacts that arise from the manner in which the globular body drapes the chemokine N-terminus within the receptor binding pocket. The chemokine N-terminus thus serves an instructive, although not solely determinative, role in dictating signaling through US28. In sum, these studies establish the plasticity of ligand-dependent US28 signaling.

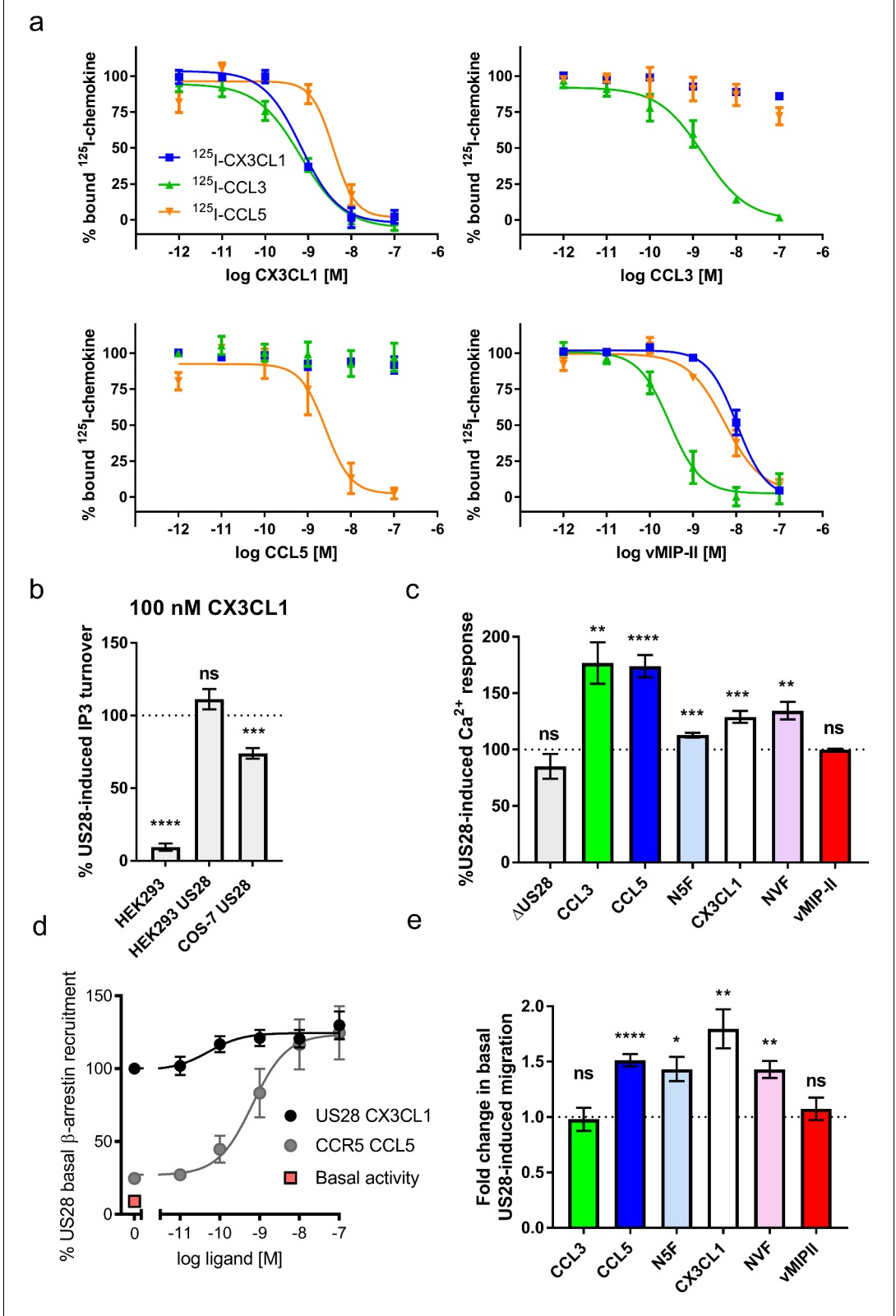

**Figure 1.** Natural and chimeric chemokines are agonists for US28. (a) Radioligand binding competition experiments with labeled CX3CL1, CCL3, and CCL5. (b) CX3CL1-induced IP3 turnover is US28 and cell type specific. Dotted line indicates US28 basal activity; all statistics are relative to this. (c) US28-induced calcium responses of 100 nM natural and chimeric chemokines. Dotted line indicates US28 basal activity; all statistics are relative to this. (d) US28 basal β-arrestin recruitment leaves narrow dynamic range in which to observe ligand effects. (e) Migration effects of 100 nM natural and chimeric

*Figure 1 continued on next page*

*Figure 1 continued*

chemkines at US28. Dotted line indicates US28 basal activity; all statistics are relative to this. All data are given as mean ± s.e.m. of at least three independent biological replicates. * p<0.05, ** p<0.01, *** p<0.001, **** p<0.0001 with respect to basal activity using one sample t-test, two-tailed.

DOI: https://doi.org/10.7554/eLife.35850.002

The following source data and figure supplements are available for figure 1:

**Source data 1.** Full statistics for wild type and chimeric chemokines.
DOI: https://doi.org/10.7554/eLife.35850.006
**Figure supplement 1.** Calcium signaling dose-response plots for wild type and chimeric chemokines.
DOI: https://doi.org/10.7554/eLife.35850.003
**Figure supplement 2.** Cell migration dose-response plots for wild type and chimeric chemokines.
DOI: https://doi.org/10.7554/eLife.35850.004
**Figure supplement 3.** vMIP-II antagonizes chemokine signaling via US28.
DOI: https://doi.org/10.7554/eLife.35850.005

## The sequence space of chemokine agonism

We next explored the structural diversity of chemokine N-terminal sequences compatible with US28 engagement to uncover sequence hallmarks that correlate with particular signaling pathways and downstream functions, such as induction of cell migration. We used a yeast-displayed library of diverse chemokine variants to map the sequence specificity of receptor engagement and activation by the Site 2 N-terminal peptide. Single chain fusions of US28 with intracellularly directed nanobodies (*Burg et al., 2015*) (Nb7 or Nb11) enable purification of stable, *apo*-US28 (*Figure 2— figure supplement 1*) that can be used to stain yeast cells displaying chemokine on their surface (*Figure 2a*), thereby solving the general problem of structural instability of GPCRs purified in *apo* form (*Handel, 2015*; *Rosenbaum et al., 2009*; *Wu et al., 2010*). The alpaca nanobodies were raised against the US28/CX3CL1 complex, and, importantly, structures of CX3CL1-bound US28 with and without Nb7 reveal no significant differences in receptor conformation. This confirms that the nanobody does not deform US28 but merely selects a subset of pre-existing, stable, active-like conformations. Thus, these nanobody fusions serve as 'molecular casts' with which to screen yeast displayed chemokine libraries using purified recombinant US28. Interestingly, we found that the two nanobodies endow US28 with different chemokine binding pharmacologies: US28Nb7 was permissive for binding of all tested chemokines, whereas US28Nb11 showed impaired binding of CCL5 and CCL3 (*Figure 2b*). This finding conforms with radioligand competition experiments and suggests that the two nanobodies are stabilizing partially non-overlapping US28 conformational subsets.

A version of US28Nb7 was created that deletes the receptor N-terminus, thereby precluding Site 1 interaction with chemokines. Surprisingly, CX3CL1 and CCL5 show virtually unchanged affinity in the absence of Site 1 interaction (*Figure 2—figure supplement 2*). This result stands in contrast to analogous experiments in the absence of nanobody (*Casarosa et al., 2005*) and suggests that nanobody7 selectively potentiates the affinity of CCL5 and CX3CL1 for US28 via Site 2, as has been observed in other GPCRs (*Staus et al., 2016*).

Having seen the importance of Site 2 for US28Nb7 engagement, a CX3CL1 library was constructed in which the seven N-terminal residues of the chemokine were randomized (*Figure 2—figure supplement 3*). This library was displayed on the surface of yeast and selected with increasing stringency for affinity to each of the nanobody-stabilized receptors (*Figure 2c*). Following selection, the remaining pools of yeast were deep-sequenced to identify CX3CL1 sequences compatible with binding either US28Nb7 or US28Nb11. This sequencing data showed that tens of thousands of unique chemokine N-termini are able to bind US28 with high affinity (*Figure 2d*). While the US28Nb11 selected chemokines appear slightly more converged, the most abundant unique variants comprise only a small fraction of the total sequence count. To visualize these broad sequence landscapes, pairs of N-termini that differ at only one of the seven randomized residues were clustered; to ensure significance we limited our analysis to sequences that appeared more than 10 times within the pool. 794 unique US28Nb7-selected N-termini met these criteria, with each connected to an average of 2.7 related sequences. Partitioning of these sequences into distinct subpopulations is not observed. After US28Nb11 selection, 11,415 unique N-termini had at least one partner that differed at only one position and each sequence connected to an average of 9.6 relatives.

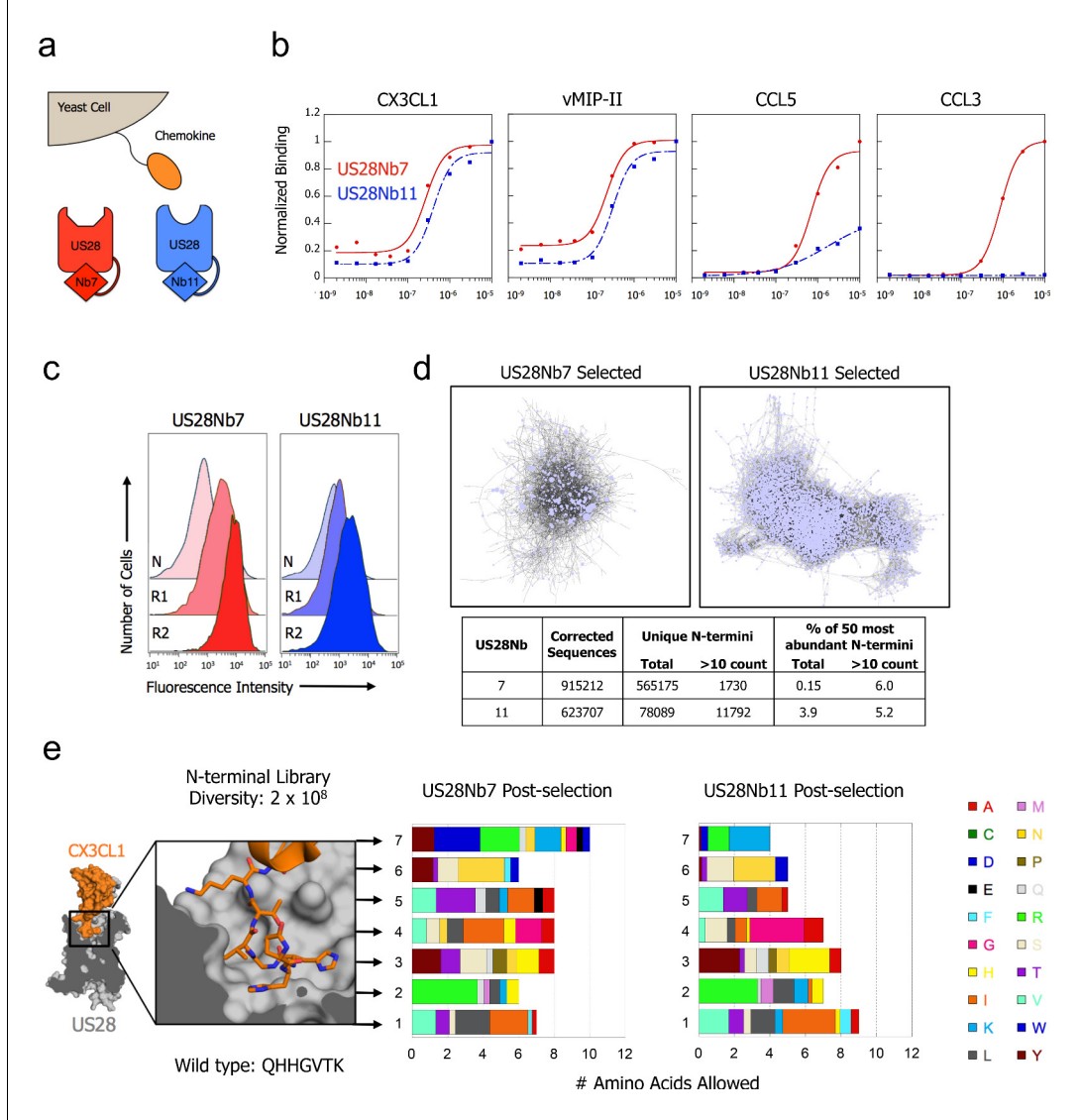

**Figure 2.** CX3CL1 library reveals chemokine sequence promiscuity and biased agonism. (**a**) Illustration of yeast-displayed chemokine and nanobody-stabilized receptors. (**b**) Effect of intracellular nanobody 7 or 11 on US28 binding to yeast-displayed chemokines. All data points are normalized to binding at 1 μM US28Nb7 (**c**) Increase in binding at 1 nM US28 after selection with either US28Nb7 or US28Nb11. (**d**) Clustering of CX3CL1 N-terminal sequences revealed by deep-sequencing after selection. Each point is a unique N-terminus and sequences sharing 6 of 7 amino acids are connected. (**e**) Degree of amino acid convergence at each position of the CX3CL1 N-terminus. Amino acids with >3% abundance after selection are considered allowed and the size of a shaded region corresponds to that amino acid's frequency (PDB: 4xt1).

DOI: https://doi.org/10.7554/eLife.35850.007

The following source data and figure supplements are available for figure 2:

**Source data 1.** Unique CX3CL1 N-termini identified by deep sequencing after selection with US28Nb7 or US28Nb11.
DOI: https://doi.org/10.7554/eLife.35850.012

**Figure supplement 1.** US28-nanobody single chain constructs enable purification of *apo*-receptor.
DOI: https://doi.org/10.7554/eLife.35850.008

**Figure supplement 2.** Effects of US28Nb7 N-terminal deletion on yeast-displayed wild-type chemokine binding.
DOI: https://doi.org/10.7554/eLife.35850.009

**Figure supplement 3.** CX3CL1 'Site 2' library design.
DOI: https://doi.org/10.7554/eLife.35850.010

**Figure supplement 4.** Effect of minimal chemokine Site 2 on US28Nb7 binding.
DOI: https://doi.org/10.7554/eLife.35850.011

A residue-by-residue analysis of the combinatorial chemokine N-termini from the library selections shows that sequence promiscuity extends along the full length of randomized positions (*Figure 2e* and *Figure 2—figure supplement 3*). Despite this diversity, CX3CL1 with a polyglycine amino terminus shows minimal binding to US28Nb7, confirming the necessity of certain N-terminal contacts (*Figure 2—figure supplement 4*). While the CX3CL1 wild type amino terminus appears in the sequencing, the wild-type identity rarely emerges among the most abundant amino acids at a given position. For example, hydrophobic amino acids enrich over the wild-type glutamine at position 1, while arginine enriches over the wild-type histidine at position 2. As the same hydrophobic amino acids score the worst in computational signal sequence cleavage site recognition, it is unlikely that this result is simply due to biases in processing or expression (*Nielsen, 2017*).

A collection of highly diverse sequences was chosen from each of the two different nanobody selections for further signaling characterization (*Figure 3a and b* and *Figure 3—figure supplements 1* and *2*). These chemokines bound US28Nb7 with affinities akin to wild-type CX3CL1 on yeast, yet revealed comparatively weak competition with wild-type CX3CL1 in radioligand binding studies (*Figure 3—figure supplement 3*). This behavior is reminiscent to that of US28's natural chemokine repertoire reported above. Because the high basal signaling of US28 results in a narrow dynamic range in which to measure differences between agonists, we can only make qualitative conclusions about signaling bias. Nevertheless, nearly all these chemokines induce some level of cell migration, though none surpass that of wild-type CX3CL1 (*Figure 3b*). Conversely, G protein activation stronger than wild-type CX3CL1 was elicited by LLPHANY (CX3CL1.35) (*Figure 3a*). To determine if the LLPHANY sequence contained particular hallmark residues that correlate with G protein signaling, homologous sequences to CX3CL1.35 that emerged from the deep sequencing were tested (*Figure 3c and d*). These sequences include variants with mutations throughout the CX3CL1.35 N-terminus. Surprisingly, despite the substantial sequence differences of these variants, most induced G protein signaling in a qualitatively similar fashion to CX3CL1.35. Similarly, none of the CX3CL1.35 family members showed significantly decreased cell migration. This result demonstrates that for a given ligand sequence motif, signaling effects are relatively insensitive to individual sequence substitutions.

## Structural basis for US28's chemokine promiscuity

We determined the crystal structure of the US28 fusion to nanobody7 in complex with the engineered chemokine CX3CL1.35 to 3.5 Å resolution (*Figure 4a*). Crystallization of the CX3CL1.35 complex by Lipidic Cubic Phase required an additional nanobody, raised by alpaca immunization against *apo*-US28Nb7 (*Figure 4—figure supplement 1a,b*). CX3CL1.35 contacts a symmetry-related nanobody B1 (*Figure 4—figure supplement 2a and b*) raising the question of whether the native conformation of the complex enables lattice contacts or, conversely, whether lattice contacts induce a non-native conformational change in the complex. The former explanation is supported by the fact that exhaustive attempts to crystallize other CX3CL1 library variants with US28Nb7 and nanobody B1 fail to yield crystals.

The chemokine globular body sits atop the receptor in a very similar disposition as wild type CX3CL1, albeit CX3CL1.35 is rotated by 16.8° (*Figure 4a* and *Figure 4—figure supplement 3*), contributing to reduced interaction between the chemokine N-loop and US28 ECL2 (*Figure 4—figure supplement 4*). This rotation is markedly greater than the 3° distortion in wild-type CX3CL1 bound to US28 in structures with and without such lattice contacts (*Burg et al., 2015*)(*Figure 4—figure supplement 2c and d*), further supporting the notion that this pose is not artefactual. The chemokine N-termini trace markedly different paths within the ligand-binding pocket of US28 (*Figure 4b*, *Figure 4—figure supplement 5*, and *Figure 4—source data 1*). The interaction chemistries and structural environments for how CX3CL1 versus CX3CL1.35 engage US28 are distinct, with each chemokine occupying both overlapping and spatially segregated regions of the US28 orthosteric pocket (*Figure 4c*). CX3CL1 forms extensive contacts with US28's minor pocket at TM1, TM2, TM3, and TM7 and ECL2 (*Figure 4e and g*). The N-terminus of CX3CL1.35 also fills the minor pocket sharing contacts with Glu277[7.39] (*Figure 4d and f*; *Figure 4—source data 1*), which is also commonly contacted by small molecule chemokine ligands (*Lückmann et al., 2017*; *Rosenkilde and Schwartz, 2006*), as well as Trp89[2.60] and Phe111[3.32] [superscripts refer to the Ballesteros-Weinstein nomenclature (*Ballesteros and Weinstein, 1995*). Chemokine interaction at these three residues is broadly conserved across receptors and strongly affects ligand affinity (*Arimont et al., 2017*). Recent shotgun mutagenesis of CXCL12 with CXCR4 also implicates these residues in the initiation

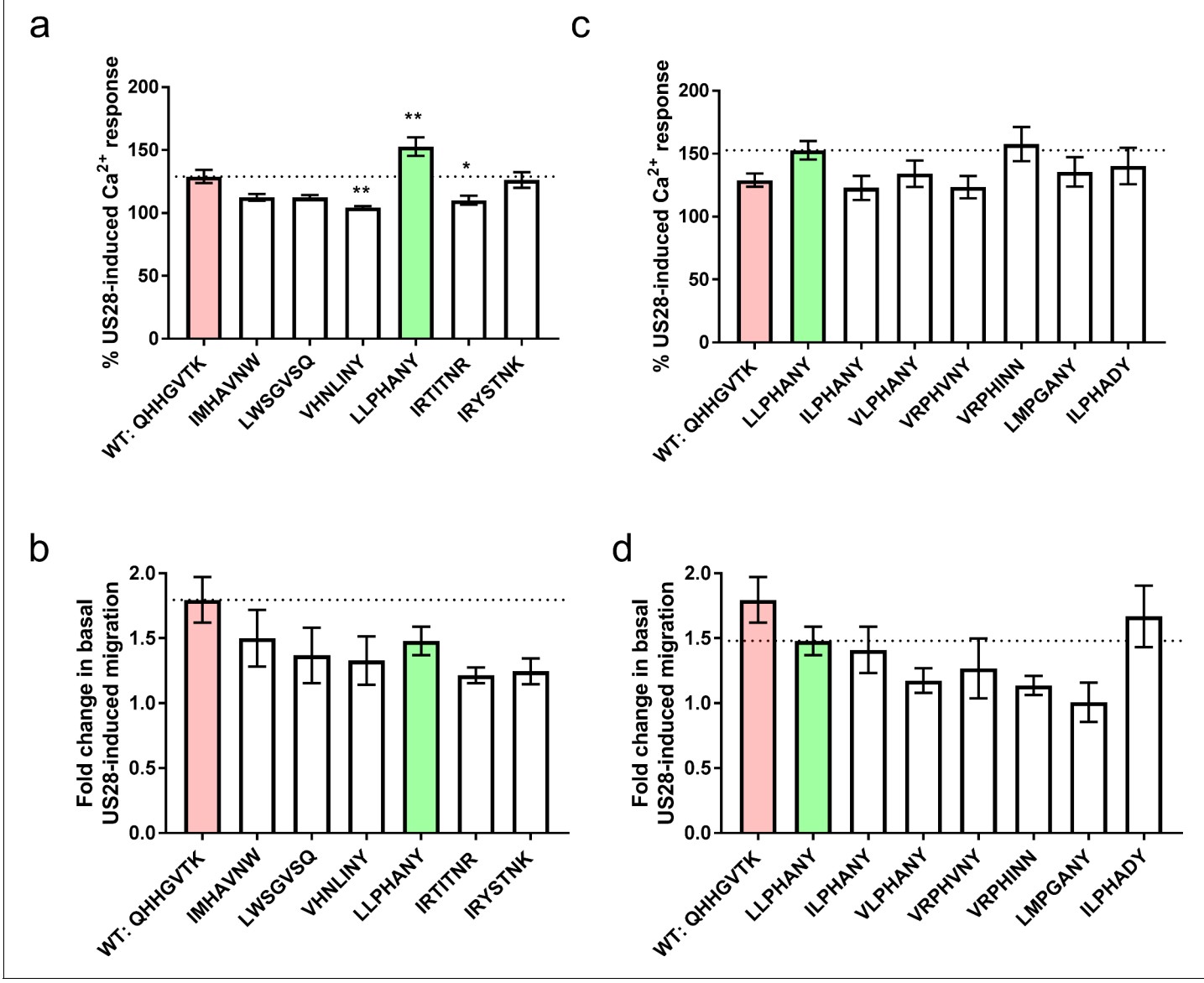

**Figure 3.** Signaling behavior or selected CX3CL1 library variants. (**a**) Calcium and, (**b**) migration responses of diverse chemokines revealed by deep sequencing at 100 nM ligand. Dotted lines indicate wild-type CX3CL1 activity (red); all statistics are relative to this. CX3CL1.35, selected for further study, is indicated in green. (**c**) Calcium and, (**d**) migration responses of CX3CL1.35-related sequences from deep sequencing at 100 nM ligand. Dotted lines indicate CX3CL1.35 (green) activity; all statistics are relative to this. All data are given as mean ± s.e.m. of at least three independent biological replicates. * $p<0.05$, ** $p<0.01$, *** $p<0.001$ by one-way ANOVA (Dunnett's test).

DOI: https://doi.org/10.7554/eLife.35850.013

The following source data and figure supplements are available for figure 3:

**Source data 1.** Full statistics for selected CX3CL1 library variants.

DOI: https://doi.org/10.7554/eLife.35850.017

**Figure supplement 1.** Calcium signaling dose-response plots for selected CX3CL1 library variants.

DOI: https://doi.org/10.7554/eLife.35850.014

**Figure supplement 2.** Cell migration dose-response plots for selected CX3CL1 library variants.

DOI: https://doi.org/10.7554/eLife.35850.015

**Figure supplement 3.** Characterization of representative CX3CL1 library variants for receptor binding.

DOI: https://doi.org/10.7554/eLife.35850.016

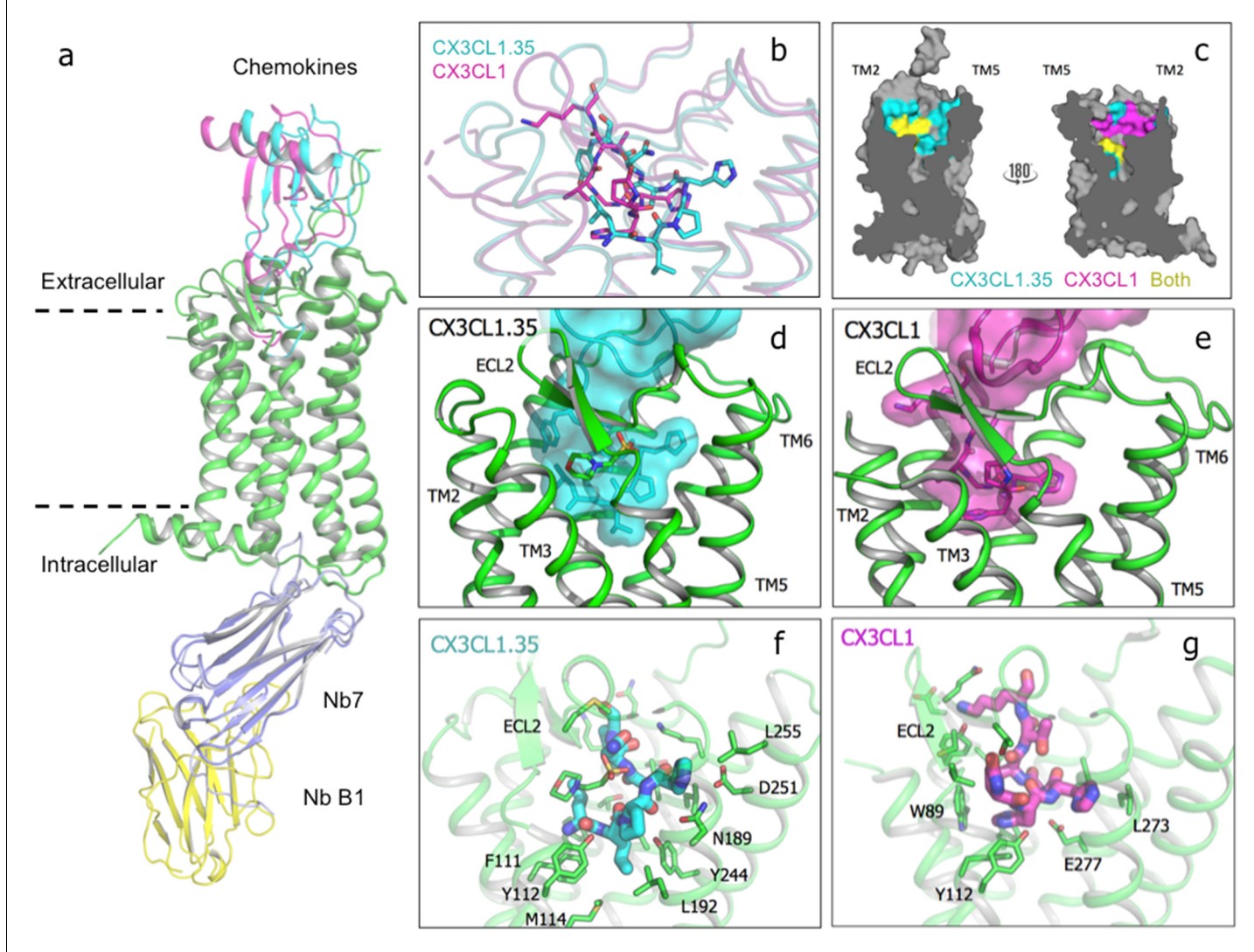

**Figure 4.** Crystal structure of US28 bound to engineered CX3CL1 and comparison with the wild-type chemokine. (a) CX3CL1.35-US28 and CX3CL1-US28 structural alignment based on US28 transmembrane helices (PDB:4xt1). (b) Wild type and engineered chemokine N-termini trace different paths in US28's binding pocket. (c) Cutaway image of US28 with chemokine contacts highlighted for CX3CL1.35 (teal), CX3CL1 (purple), or both (yellow). (d) CX3CL1.35 fills the entire binding pocket. (e) CX3CL1 hugs the TM2 side of US28. (f) US28 side chains contacted by CX3CL1.35 in the receptor binding pocket. (g) US28 side chains contacted by CX3CL1 in the receptor-binding pocket.

DOI: https://doi.org/10.7554/eLife.35850.018

The following source data and figure supplements are available for figure 4:

**Source data 1.** CX3CL1.35 interactions with US28Nb7.
DOI: https://doi.org/10.7554/eLife.35850.024

**Figure supplement 1.** Nanobody B1 selections and binding interface.
DOI: https://doi.org/10.7554/eLife.35850.019

**Figure supplement 2.** Crystal packing of CX3CL1.35-US28Nb7-nanobody B1 complex structure.
DOI: https://doi.org/10.7554/eLife.35850.020

**Figure supplement 3.** US28 structural comparisons.
DOI: https://doi.org/10.7554/eLife.35850.021

**Figure supplement 4.** CX3CL1.35 'Site 1.5' ECL2 contacts.
DOI: https://doi.org/10.7554/eLife.35850.022

**Figure supplement 5.** CX3CL1.35 'Site 2' and receptor contacts.
DOI: https://doi.org/10.7554/eLife.35850.023

of receptor signaling (*Wescott et al., 2016*). Unlike the wild-type chemokine, CX3CL1.35 also projects into the major pocket of the receptor toward TM5 and TM6, contacting Asn189$^{5.39}$, Leu192$^{5.42}$, Tyr244$^{6.51}$, Leu248$^{6.55}$, and Asp251$^{6.58}$ (*Figure 4d and f*). These residues superficially surround the Trp241$^{6.48}$ rotamer switch that is implicated in the transition between inactive and active GPCR structures (*Arimont et al., 2017*; *Latorraca et al., 2017*). CX3CL1.35 loses the extensive direct contacts to TM2 and ECL2 evident in the structure of the wild-type chemokine (*Figure 4b* and *Figure 4—figure supplement 4*). Indirect contacts between CX3CL1.35 and US28 ECL2 are mediated by the sulfonate moiety of a MES ion from the crystallization buffer (*Figure 4b*).

Re-examining the signaling activity of the specific CX3CL1.35 sequence family members (*Figure 3c and d*) in light of this structure demonstrates that signaling for either pathway is largely unaffected by even drastic changes to the interactions within the receptor binding cavity. Calcium signaling and migration are robust to disruption of receptor contacts throughout the binding pocket suggesting either that specific side chain contacts are largely unimportant or that significant conformational rearrangements of the chemokine amino terminus are made to preserve contacts. The most prosaic explanation for the relative lack of sequence selectivity imposed by the US28 pocket, yet ability of unrelated chemokine sequences to elicit differential signaling outputs, would be that the steric bulk of the ligand is more important than specific bonding chemistries. If steric bulk, which would apply strain to the walls of the US28 binding pocket, were the principal determinant of signaling output, an almost unlimited number of sequences could elicit similar signaling outputs, which is consistent with our data.

## Extracellular rearrangements upon chemokine binding

To assess the impact of ligand binding on the structure and dimensions of the US28 ligand binding pocket, we determined the crystal structure of *apo*-US28Nb7 without an extracellular ligand to 3.5 Å resolution (*Figure 5—figure supplement 1a*). Together, the CX3CL1-bound, CX3CL1.35-bound, and *apo*-US28 structures allow direct comparisons among the basal and qualitatively β-arrestin- and G-protein-biased states of the same GPCR, with the caveat that the bound nanobodies prevent us from reaching conclusions about structural differences in the intracellular regions of US28 (*Figure 5—figure supplement 1b*). In contrast to the chemokine-bound structures, access to the extracellular binding cavity of US28 is significantly constricted in *apo*-US28, where ECL1 and ECL2 collapse inward (*Figure 5a*). Each chemokine elicits distinct conformational changes in US28 in order to gain access to the binding pocket. CX3CL1 displaces ECL1 and ECL2 away from the receptor core, uncovering the binding pocket but causing minimal overall distortion (*Figure 5b*). Conversely, CX3CL1.35 distorts TM1, TM6, and TM7 away from the receptor core, resulting in an expanded binding pocket.

By way of comparison, small molecule agonists induce subtle contraction of the binding pocket, drawing in TM5 (as in the ß2-adrenergic receptor [*Rasmussen et al., 2011*; *Ring et al., 2013*]), TM6 (as in the M2 muscarinic receptor [*Kruse et al., 2013*]), or TM3 (as in the μ-opiod receptor [*Huang et al., 2015*]) (*Latorraca et al., 2017*). Recent structures of endothelin B (ETb) suggest that peptide agonists may behave more like chemokines. *Apo* and ET1-bound structures show that activation results in large inward translations of the extracellular ends of TM6 and TM7 toward TM3 (*Shihoya et al., 2016*).

The pronounced differences in ECL conformation and contacts with chemokine raise the possibility that chemokines may function as bitopic ligands via site 1.5 and site 2 interactions, with both interaction sites working in concert to determine a chemokine's precise signaling profile. This notion is supported by the unique signaling profiles of chimeric chemokines NVF and N5F. ECL stabilization is a common mechanism of allosteric modulation among group A GPCRs (*Christopoulos, 2014*; *Kruse et al., 2013*) and recent mutagenesis studies suggest a role for ECL2 in chemokine receptor signaling (*Ziarek et al., 2017*). Indeed, toggling ligand interactions between ECL2 and TM5 has recently been proposed as a general mechanism of inducing signaling bias within aminergic class A GPCRs (*McCorvy et al., 2018*).

## Conclusions

Our approach using chemokine libraries has revealed that US28 can accommodate at least thousands of chemically diverse chemokine N-termini, perhaps many more, indicating an astonishing

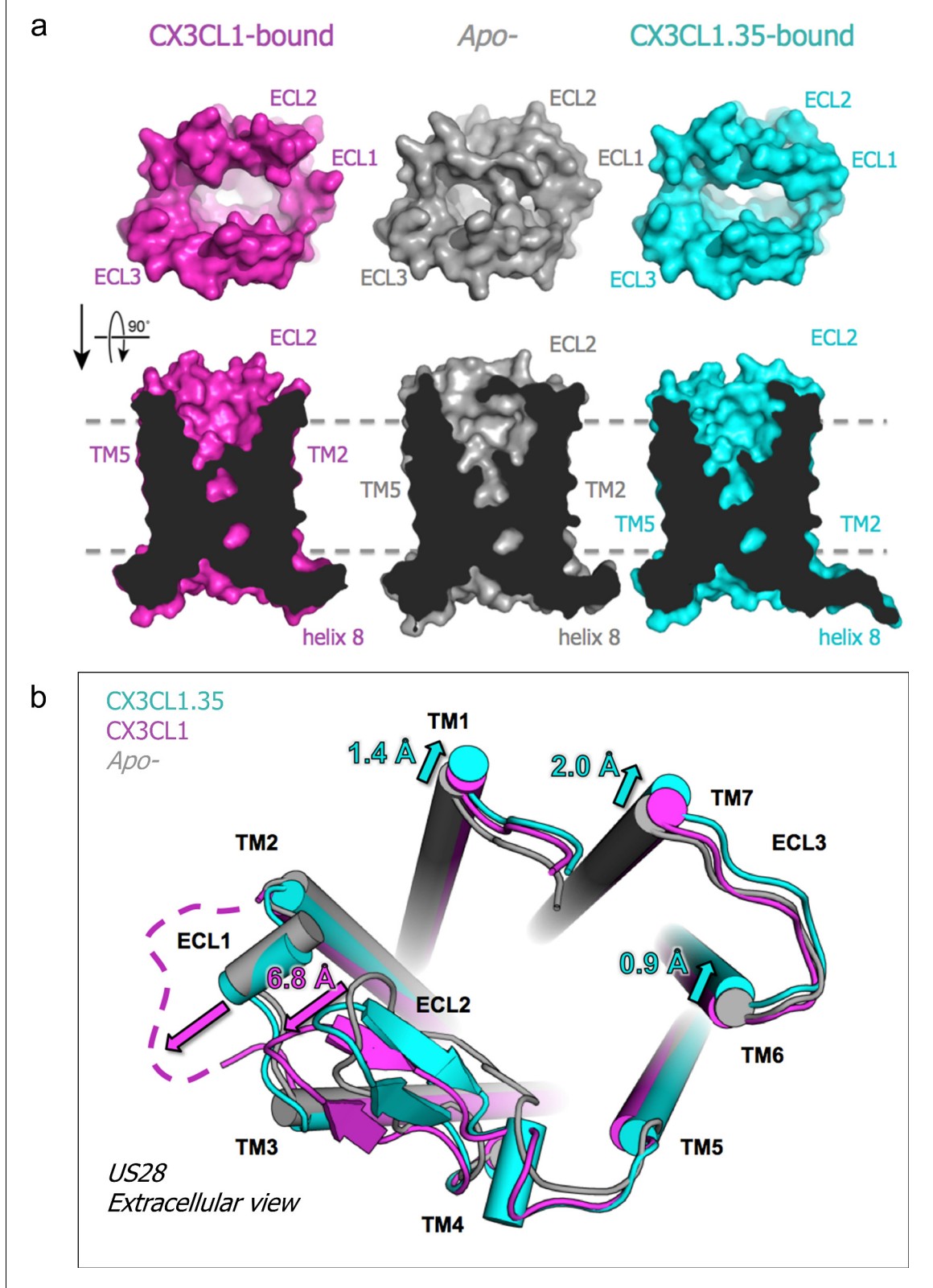

**Figure 5.** Chemokine-dependent conformational changes in US28. (a) Top-down and side views of *apo-*, CX3CL1-, and CX3CL1.35-bound US28 showing expanded access to the receptor binding pocket when chemokine is present. (b) Structural alignment based on US28 transmembrane helices showing unique US28 conformational changes caused by each chemokine. (PDB: 4xt1). Crystallographic data and refinement statistics are summarized in *Figure 5—source data 1*.

*Figure 5 continued on next page*

*Figure 5 continued*

DOI: https://doi.org/10.7554/eLife.35850.025

The following source data and figure supplement are available for figure 5:

**Source data 1.** Summary of crystallization and crystallographic statistics.

DOI: https://doi.org/10.7554/eLife.35850.027

**Figure supplement 1.** *Apo-* structure and comparison of intracellular faces.

DOI: https://doi.org/10.7554/eLife.35850.026

degree of ligand cross-reactivity for a cell surface receptor. Without apparent sequence patterns, these engineered chemokines induce the full spectrum of signaling bias at US28. Natural chemokine sequences merely represent a subset of local minima in a broad signaling fitness landscape. The combined structural and deep sequencing data suggest that G protein activation is elicited as a consequence of steric bulk distorting the walls the major pocket of the receptor, dilating the receptor's extracellular face, rather than highly specific and sequence-specific bonding chemistries. This relative sequence-insensitive mechanism, which is highly unusual for protein-protein interactions, is likely enabled by the N-terminal chemokine sequence binding within a capsule-shaped US28 pocket, affording numerous opportunities for adventitious bonding interactions between the walls of the pocket and alternative conformations of the N-terminal peptide. This is as opposed to the generally broad and exposed interfaces seen in protein-protein interactions, which are generally much less tolerant to substitution. The properties exhibited by US28 explain how the virus evolved a human GPCR to be highly promiscuous in order to promote viral survival and offer the broader possibility that GPCR signaling can be activated by surrogate agonists that are unrelated to the natural ligands and activate new signaling pathways.

# Materials and methods

## Key resources table

| Reagent type (species) or resource | Designation | Source or reference | Identifiers | Additional information |
|---|---|---|---|---|
| Gene (human cytomegalovirus) | US28 (unique short region) | PMID 7961796 | | strain TOWNE |
| Strain, strain background (*Saccharomyces cerevisiae*) | EBY100 | Gift from Prof. Dane Wittrup (PMID 17406305) | | Yeast cells |
| Cell line (Spodoptera frugiperda) | SF9 | ATCC | CTL-1711 | Insect cells used for baculovirus production |
| Cell line (Trichoplusia ni) | Hi5 | Invitrogen | BTI-TN-5B1-4 | Insect cells used for baculovirus expression of NbB1 |
| Cell line (*Homo sapiens*) | HEK293 | ATCC | CRL-1573 | Mammalian cells used for Ca$^{2+}$ signaling assay |
| Cell line (*Homo sapiens*) | HEK293-US28 wt | PMID 23303826 | | Mammalian cells used for Ca$^{2+}$ signaling assay and IP3 assays |
| Cell line (*Homo sapiens*) | HEK293S GnTI- | Gift from Prof. H. Gobind Khorana (PMID 12370423) | | Mammalian cells used for baculovirus expression of US28 variants and chemokines |
| Cell line (*Homo sapiens*) | Flp-In TREx 293 | Invitrogen | R78007 | Hamster cells used for β-arrestin assay |
| Cell line (Chinese hamster ovary) | CHO-K1 EA-arrestin | DiscoverixRx | 93–0164 | |
| Transfected construct (β-arrestin recruitment) | US28 wt/ProLink/b-galactose | This report | | Vector provided by DiscoverixRx |

*Continued on next page*

*Continued*

| Reagent type (species) or resource | Designation | Source or reference | Identifiers | Additional information |
|---|---|---|---|---|
| Antibody | Anti-protein C (mouse IgG1) | ATCC | HB-9892 | Antibody used for staining yeast bound to protein C tagged target proteins. Purified from HPC-4 MOUSE HYBRIDOMA for Alexa647 labeling. |
| Antibody | Anti-FLAG M1 (mouse IgG2a) | Gift from Prof. Brian Kobilka (PMID 17962520) | | Antibody used for US28 purification. Purified from M1 HYBRIDOMA to prepare anti-FLAG M1 affinity resin |
| Antibody | Myc-Tag (9B11) Mouse mAb (Alexa Fluor 488 Conjugate) | Cell Signaling Technology | 2279 | Antibody used for staining yeast properly displaying proteins of interest with Myc-tag |
| Recombinant DNA reagent | BestBac Linearized Baculovirus DNA 2.0, Expression Systems, 91–002 | Expression Systems | 554739 | |
| Peptide, recombinant protein | CCL3 | Peprotech | 300–08 | |
| Peptide, recombinant protein | CCL5 | Peprotech | 300–06 | |
| Peptide, recombinant protein | CX3CL1 | Peprotech | 300–31 | |
| Commercial assay or kit | PathHunter β-arrestin assay | DiscoverixRx | | |
| Commercial assay or kit | MiSeq v2 2 × 150 | Illumina | MS-102–2002 | |
| Commercial assay or kit | MiSeq v2 2 × 250 | Illumina | MS-102–2003 | |
| Chemical compound, drug | Alexa Fluor 647 NHS Ester (Succinimidyl Ester) | Thermo Fisher Scientific | A37573 | Labeling reagent for anti-protein C antibody |
| Chemical compound, drug | monoolein | Sigma | M7765 | For LCP |
| Chemical compound, drug | cholesterol hemisuccinate tris salt | Anatrace | CH210 | For membrane protein purification and yeast staining buffer |
| Chemical compound, drug | cholesterol | Sigma | C8667 | For LCP |
| Chemical compound, drug | n-dodecyl-β-D-maltoside | Anatrace | D310 | For membrane protein SEC buffer |
| Chemical compound, drug | n-dodecyl-β-D-maltoside | Anatrace | D310S | For membrane protein solubilization buffer, affinity column buffer, and yeast staining buffer |
| Software, algorithm | Prism7 | GraphPad | | |
| Software, algorithm | XDS | PMID 20124692 | | Data integration, scaling, space-group assignment and merging |
| Software, algorithm | Phaser | PMID 19461840 | | Molecular replacement |
| Software, algorithm | Phenix suite | PMID 20124702 | | Structure refinement |
| Software, algorithm | Coot | PMID 20383002 | | Structural model building |
| Software, algorithm | PyMol | Schrödinger | | Structural visualization/ figure preparation |
| Software, algorithm | Pandaseq | PMID 22333067 | | |

*Continued*

| Reagent type (species) or resource | Designation | Source or reference | Identifiers | Additional information |
|---|---|---|---|---|
| Software, algorithm | Geneious | Biomatters | | |
| Software, algorithm | Matlab | Mathworks | | |
| Software, algorithm | Cytoscape | PMID 14597658 | | Cluster analysis visualization |
| Software, algorithm | Cytobank | Cytobank, Inc. | | Flow cytomettry visualization |
| Software, algorithm | KaleidaGraph | Synergy Software | | |
| Software, algorithm | Clustering algorithm | PMID 24855945 | | |
| Other | CNBr-Activated Sepharose 4 Fast Flow | GE Healthcare | 17098101 | |
| Other | LS columns | Miltenyi | 130-042-401 | |
| Other | Anti-Cy5/Anti-Alexa Fluor 647 MicroBeads | Miltenyi | 130-091-395 | |
| Other | MidiMACS Magnetic Separator | Miltenyi | 130-042-302 | |

## Design and purification of US28-nanobody fusions

US28 was truncated by 10 amino acids at the N-terminus and 44 amino acids at the C-terminus. Examination of the CX3CL1-US28-nanobody7 structure (PDB: 4xt1) allowed for the design of a linker between the C-terminus of US28ΔNΔC and the N-terminus of nanobody7 composed of two thrombin recognition sites and a six residue Gly-Ser linker. This construct, termed US28Nb7, was decorated with HA signal peptide, an N-terminal FLAG epitope tag and a C-terminal 3C protease site, as well as C-terminal protein C and 8xHis tags. The same design was used to make a single-chain construct between US28 and nanobody11 (previously demonstrated to be competitive with nanobody7 [*Burg et al., 2015*]), termed US28Nb11.

US28Nb7 and US28Nb11 were expressed in HEK293S GnTI- cells using BacMam baculovirus transduction. Baculovirus was added to cells at a density of $2 \times 10^6$ cells ml$^{-1}$ and culture bottles were shaken for 24 hr at 37°C with 5% $CO_2$. After harvesting, cells were washed with PBS supplemented with 5 mM EDTA and 1:1000 protease inhibitor cocktail (PIC, Sigma Aldrich, St. Louis MO) and stored at −20°C. Cell pellets were thawed and lysed with a Dounce homogenizer in a solution composed of 20 mM Tris-Cl pH 8.0, 5 mM EDTA, 2 mg ml$^{-1}$ iodoacetamide, and 1:1000 PIC. The lysate was centrifuged at 40,000 x $g$ for 1 hr and the membrane pellet was resuspended and rotated for 2 hr in a solubilization buffer consisting of 10 mM HEPES pH 7.4, 150 mM NaCl (HBS), 1% (w/v) dodecylmaltoside (DDM), 0.2% (w/v) cholesterol hemisuccinate (CHS), 10% (v/v) glycerol, 2 mg ml$^{-1}$ iodoacetamide, and cOmplete PIC (Roche, Basel Switzerland).

After centrifugation, 5 ml Ni-NTA resin (Qiagen, Hilden Germany ) per L of initial culture was added and stirred at 4°C overnight. The resin was then collected in a column; washed with HBS with added 0.1% (w/v) DDM, 0.02% (w/v) CHS, 10% (v/v) glycerol, and 20 mM imidazole (wash buffer); and eluted in wash buffer supplemented to 200 mM imidazole. The eluate was then adjusted to 2 mM CaCl$_2$ and further purified over an anti-FLAG M1 affinity column. The receptor was eluted with 0.2 mg ml$^{-1}$ FLAG peptide and 5 mM EDTA and further purified by size exclusion chromatography using a buffer containing HBS, 0.02% (w/v) DDM and 0.004% (w/v) CHS.

US28Nb7 and US28Nb11 'competitor' constructs (used in kinetic selections) were digested with 1:10 (w/w) 3C protease for 2 hr at room temperature to remove the protein C epitope tag prior to size exclusion chromatography. In ΔN US28Nb7 (used in nanobody selections and chemokine-binding assay (*Figure 2—figure supplement 2*)), the thrombin sites in the linker between US28 and nanobody7 were replaced with Gly-Ser linkers of equal length and a thrombin site was introduced with a three amino acid Gly-Ser linker N-terminal to Cys22 of US28. This construct was purified as

above, digested with 1:100 (w/w) thrombin at 4°C overnight and cleared over an anti-FLAG M1 affinity column prior to size exclusion chromatography. In all cases, the final protein was concentrated, aliquoted, and stored frozen before staining experiments.

## Creation and staining of yeast display constructs

Chemokines were synthesized as N-terminal fusions to the yeast surface protein Aga2p (IDT, San Jose CA). Constructs were cloned into the vector pYAL with the Aga2p leader sequence, leaving the chemokine N-terminus free. The plasmid vector contains a Gly-Ser linker and a Myc epitope tag between the chemokine and Aga2p. Constructs were then electroporated into electrocompetent EBY-100 yeast, passaged in synthetic defined medium (SDCAA) and protein expression was induced in SGCAA pH 4.5 media at 20°C for 24–60 hr as described (*Chao et al., 2006*), until maximum Myc epitope tag staining was observed (typically 40–70% of total population).

A CX3CL1 N-terminal library was prepared by assembly PCR with oligonucleotide primers containing degenerate codons at the first seven amino acids of the chemokine (contiguous with the first cysteine) (*Figure 2—figure supplement 2*). The amplicon contained 50 base pairs of homology to pYAL. The mutagenic CX3CL1 DNA and linearized pYAL vector were co-electroporated into EBY100 yeast to yield a library with $2.0 \times 10^8$ transformants.

Induced yeast displaying chemokine were washed with HBS with added 2 mM $CaCl_2$, 10 mM maltose, 0.1% (w/v) BSA, 0.02% (w/v) DDM and 0.004% (w/v) CHS (Staining Buffer) and stained with varying concentrations of the desired receptor construct for 2 hr at 4°C. The yeast were then washed with staining buffer and stained with Alexa-647 conjugated anti-protein C antibody and Alexa-488 conjugated anti-Myc antibody (Cell Signaling, Danvers MA) for 15 min at 4°C. After a final wash, mean cell fluorescence was measured using FL-1 and FL-4 channels of an Accuri C6 flow cytometer (BD Biosciences, Franklin Lakes NJ).

## Chemokine selection by yeast surface display

For the first round of selection, $2.0 \times 10^9$ yeast (10x the library diversity to ensure full coverage) induced with SGCAA medium were washed with staining buffer and negatively selected to eliminate any yeast clones against selecting reagents and magnetic column. Yeast were stained with Alexa-647 conjugated anti-protein C antibody for 15 min at 4°C, washed with staining buffer and magnetically labeled with 50 μl anti-Alexa-647 microbeads (Miltenyi, Bergish Gladbach Germany) in staining buffer for 15 min at 4°C. Yeast were again washed with staining buffer and unlabeled yeast were isolated by clearing through an LS column (Miltenyi) pre-equilibrated with staining buffer. These cleared yeast were then washed with staining buffer and stained with 3 nM US28Nb7 or 30 nM US28Nb11 for 2 hr at 4°C. These concentrations were chosen so that ~10% of the yeast would be bound. The yeast were then washed with staining buffer and stained with Alexa-647 conjugated anti-protein C antibody for 15 min at 4°C. Yeast were washed again with staining buffer and magnetically labeled with 250 μl anti-Alexa-647 microbeads (Miltenyi) in staining buffer for 15 min at 4°C. Yeast were again washed with staining buffer and labeled yeast were isolated by magnetic selection with an LS column (Miltenyi) pre-equillibrated with staining buffer. Magnetically sorted yeast were resuspended in SDCAA and cultured at 30°C.

A second round of magnetic selection was performed for US28Nb11 selected yeast at a concentration of 3 nM target (~10% of post-R1 yeast were now bound at this concentration) so that the same stringency of selection was now achieved for US28Nb11 as for the first round of US28Nb7 selections. For the final round of selection, a kinetic selection was performed to isolate clones with the slowest off-rates. Yeast equaling 10x the post-selection diversity after the previous round were induced in SGCAA medium. Non-specific antibody binders were again cleared as described above and the yeast were again stained with 3 nM US28Nb7 or US28Nb11 (matching the previous round) for 2 hr at 4°C. Following this, the yeast were washed with staining buffer and resuspended in 3 μM of the nanobody-matched 'competitor' construct that lacks a protein C tag (and thus will be dark in the selection). The yeast were incubated at room temperature for 90 min, after which time they were washed in staining buffer and stained with Alexa-647 conjugated anti-protein C and Alexa-488 anti-Myc antibodies for 15 min at 4°C. Yeast were washed with staining buffer and Alexa-647 positive yeast with the highest Alexa-647/Alexa-488 ratios were purified using a FACS Jazz cell sorter (BD Biosciences). Post-sorted yeast were resuspended in SDCAA medium and cultured at 30°C.

Chemokine cDNA was prepared from each of the post selection library samples, transformed into *E. coli* and sequenced, to confirm sequence convergence.

## Deep sequencing of chemokine libraries

Deep sequencing was performed as previously described (*Birnbaum et al., 2014*). Briefly, pooled plasmids from $2 \times 10^7$ yeast from each round of selection were isolated via yeast miniprep (Zymoprep II kit, Zymo Research, Irvine CA) and used as PCR template to prepare Illumina samples. Amplicon libraries were designed as follows: Illumina P5-Truseq read 1-($N_8$)-Barcode-Chemokine-($N_8$)-Truseq read 2-IlluminaP7. $N_8$ was added immediately after both sequencing primers to generate diversity for low-complexity sequencing reads. The adaptor and barcode sequences were appended via nested cycles of PCR of the purified plasmids using Phusion polymerase (NEB, Ipswich MA). The number of cycles for each round of PCR was determined by quantitation on a Bioanalyzer (Agilent, Santa Clara CA) to protect against over-amplification. Primers were proximal to the library region of the chemokine to ensure high-quality sequence reads with double coverage. Final PCR products were run on a 2% agarose gel and purified via gel extraction (QIAGEN). Purified PCR products were then quantitated, normalized for each barcoded round of selection to be equally represented, doped with 5–50% PhiX DNA to ensure sufficient sequence diversity for high-quality sequence reads and run on an Illumina Miseq with $2 \times 150$ nt Paired End reads (Illumina, San Diego CA).

To analyze the sequence data, contigs were generated for each paired end read using PANDA-seq software. The contigs were then deconvoluted into individual rounds of selection and trimmed to the chemokine sequence using Geneious version 6. The numbers of reads for each unique sequence were then summed. Sequences were then translated into peptides and any reads that contained stop codons, frameshifts, or mutations outside the library design were omitted from further analysis. Amino acid frequencies were then calculated for those unique sequences that appear with 10 or greater counts as previously described. Clustering of selected CX3CL1 amino termini was performed for US28Nb7 and US28Nb11 samples after the final kinetic selection for all sequences appearing greater than 10 copies and connected by a Hamming distance of 6 (1 position variable allowed) into a network.

## Chemokine purification

Individual CX3CL1 library variants (CX3CL1.##) selected for signaling analysis were designed and expressed as reported previously. Constructs comprised the natural CX3CL1 signal peptide, library amino acid residues 1–7, and CX3CL1 residues 8–77 under the control of the CMV promoter in the vector pVLAD6. The C-terminal mucin-like stalk was replaced with a flexible linker (SGSGSAAA) followed by a 3C protease site (LEVLFQGP) and human Fc.

CX3CL1.##−3C-Fc were expressed in HEK293S GnTI- cells with BacMam baculovirus transduction. Baculovirus was added to the cells at a density of $2 \times 10^6$ cells ml$^{-1}$ and culture bottles were shaken for 72 hr at 37°C with 5% $CO_2$. Cells were removed by centrifugation, and the culture supernatant was filtered and then stirred at 4°C overnight with 3 ml Ni-NTA resin (Qiagen) per L of supernatant. The Ni-NTA was then collected by filtration, washed with 20 mM imidazole HBS, and eluted with 200 mM imidazole HBS. To liberate the chemokine from the Fc, 1:50 (w/w) 3C protease was added to the eluate and incubated at 4°C overnight. The sample was then diluted 10x with HBS to 20 mM imidazole and run twice through a Ni-NTA column to clear the Fc and protease. The flow-through containing the final protein was concentrated, aliquoted, and stored frozen before signaling experiments.

## Cell lines

HEK293 cells were obtained from ATCC (Manassas VA) (CRL-1573), CHO-K1 EA-arrestin cell line from DiscoverX (Fremont CA) and the Flp-In TREx 293 cell line from Invitrogen (Carlsbad CA). Hi5 cells were from Invitrogen (BTI-TN-5B1-4), SF9 cells were from ATCC (CTL-1711), HEK293GnTI- cells were provided by Prof. H.G. Khorana, and EBY100 yeast cells were provided by Prof. Dane Wittrup. Cell line authentication was guaranteed by the sources where the cells were bought. All eukaryotic cell lines used for signaling and functional assays were tested negative for mycoplasma on a regular basis, before and during tissue culture.

## Cell culture, US28 constructs, and data analysis for signaling assays

HEK293 cells were grown in Dulbecco´s modified Eagle´s Medium (DMEM) supplemented with 10% fetal bovine serum (v/v), 180 units $mL^{-1}$ penicillin and 45 µg $mL^{-1}$ streptomycin. The CHO-K1 EA-arrestin2 cells were grown in Ham´s F-12 medium containing 10% fetal bovine serum, 2 mM glutamine, 180 units $mL^{-1}$ penicillin, 45 µg $mL^{-1}$ streptomycin and 250 µg $mL^{-1}$ hygromycin. The stable clones of inducible US28wt- (parental virus strain TOWNE) HEK293 cells were generated previously described (Hjortø et al., 2013) and were grown as the following the manufacturer's protocol for maintenance of parental and inducible clones (flp-in-t-Rex system, Invitrogen).

The ligands were tested in at least three individual biological replicates, each with at least two technical replicates. Three biological independent experiments was considered as the minimum required to see if the results were internally consistent. If the results from one experiment were not consistent with the two other experiments, the experiment was repeated to gain a minimum of three independent experiments showing similar results. All experiments were included in the final sum, except for cases where the controls indicated experimental problems. Experiments were excluded if the controls indicated experimental problems for example negative data for the positive controls. Data points were excluded when there was a technical mistake during the procedure of the experiment.

## Radioligand competition binding assay

Stable inducible clones of the US28-cells were seeded in poly-D-lysine (Invitrogen) coated 96-well tissue culture plates (clear plates, Costar). The number of cells seeded was determined by the apparent expression of receptors and was aimed at obtaining 5–10% binding of the added radioactive ligand. US28 receptor expression was induced by tetracycline one day after seeding the cells (0.25 µg $mL^{-1}$; Invitrogen). 48 hr post seeding, cells were washed twice in ice-cold binding buffer (50 mM HEPES pH 7.4, supplemented with 1 mM $CaCl_2$, 5 mM $MgCl_2$ and 0.5% (w/v) bovine serum albumin) and assayed by competition binding for 3 hr at 4°C using 10–15 pM $^{125}I$-CX3CL1, $^{125}I$-CCL5, or $^{125}I$-CCL3 as well as unlabeled ligand (10 pM to 100 nM in binding buffer). After incubation, cells were washed twice in ice-cold binding buffer, supplemented with 0.5 M NaCl.

## $Ca^{2+}$ mobilization assay

US28- and HEK293 cells (as a negative control) were seeded at $2 \times 10^4$ cells per well in poly-D-lysine coated 96-well plates (black, clear bottom, Costar). US28-expression was induced 1 day after seeding by addition of 0.25 µg $ml^{-1}$ tetracycline. After 24 hr, cells were loaded for 1 hr at 37°C (5% $CO_2$) in the dark with 0.2% Fluo-4 (Invitrogen) in loading buffer (19.6 mM HEPES pH 7.4, 1.25 mM probenecid, 1 mM $CaCl_2$, and 1 mM $MgCl_2$). After 1 hr incubation, cells were washed in pre-warmed loading buffer and 100 µl of the loading buffer was added to each well as a final volume. Intracellular $Ca^{2+}$ mobilization was monitored upon stimulation with various concentrations of chemokines at 37°C as fluorescence at excitation and emission wavelengths of 485 and 520 nm, respectively. The measurements were performed using a FlexStation3 (Molecular Devices, San Jose CA). If used, the antagonist was incubated in 100 µl loading buffer 10 min prior to the addition of the agonist.

## β-arrestin2 recruitment assay

The recruitment of β-arrestin2 was measured using the PathHunter β-arrestin assay (DiscoverX) as described previously (Daugvilaite et al., 2017). Briefy, cDNA encoding US28wt was fused to the ProLink C-terminal protein tag and the small fragment ofβ-galactose and cloned into pcDNA3.1+. Assays were performed using the CHO-K1 EA-arrestin cell line with the stable expression of β-galactosidase coupled to the β-gal large fragment. Cells were seeded at $2 \times 10^4$ cells per well and transfected the next day with 50 ng DNA using Fugene6 reagent (0.15 µl per well, Promega, Madison WI). 48 hr post transfection cells were stimulated with various concentrations of the chemokine for 90 min (as positive control CCR5 transient transfected cells were included, stimulated with CCL5). β-arrestin2 recruitment was detected as β-gal activity using the PathHunter detection kit (DiscoverX). Chemiluminescent substrate composed of Galacton Star Substrate, Emerald II Solution and PathHunter Cell Assay Buffer in a ratio of 1:5:19, respectively, was added to the cells (50 µL per well). The luminescent signal was determined after 60 min incubation at ambient temperature using the EnVision Multilabel Plate Reader (PerkinElmer, Waltham MA).

## Cell migration assay

Migration of serum-starved (2 hr) tetracycline induced (16 hr) US28-HEK293 was assessed using Transwell membranes (Costar; 8 µm pore size). Filters were coated for 30 min at room temperature on the lower side with 10 µg mL$^{-1}$ fibronectin (dissolved in PBS; Sigma). The fibronectin solution was removed and the filters rinsed once with PBS before they were allowed to air-dry. The filters were placed in a 24-well dish that contained low serum (0.2%) DMEM supplemented with agonists or control buffer. US28-HEK293 cells suspended in 0.2% serum DMEM (with tetracycline) were added to the upper chamber (1 × 10$^5$ cells per well). Cells were allowed to migrate for 6 hr at 37°C. Non-migrated cells were removed from the top filter surface with a damp cotton swab. Migrated cells, attached to the bottom surface, were fixed in 3.7% formaldehyde (in PBS) and dyed with crystal violet. Transmigrated cells were counted in five predefined areas on the membrane using a converted light microscope (20x objective; Leica, Wetzlar Germany).

## Alpaca immunization

A 3-year-old male alpaca (*Lama pacos*) was maintained in pasture, and immunized following a protocol authorized by the Camelid Immunogenics (Belchertown, MA) and MIT IACUC committees. The alpaca was immunized by subcutaneous injection of a 1:1 mixture of Imject alum (Thermo Scientific, Waltham MA) and 800 µg of recombinant US28Nb7 reconstituted in phospholipid vesicles. After a total of four injections (200 µg each) spaced 2 weeks apart, 50 mL of blood was harvested by venipuncture and collected into heparinized tubes.

## Alpaca nanobody yeast display library construction

Peripheral blood lymphocytes were isolated from total blood by Ficoll-Paque gradients (Ficoll-Paque Plus, GE Healthcare, Little Chalfont United Kingdom). Total RNA was isolated from ~2×10$^6$ fresh peripheral blood lymphocytes (PBLs) using the RNeasy Plus Mini Kit (QIAGEN), following the manufacturer's guidelines. First strand cDNA synthesis was performed using SuperScript III reverse transcriptase (Life Technologies, Carlsbad CA) and a combination of oligo dT, random hexamer or the immunoglobulin-specific primers AlCH2 and AlCH2.2 (*Maass et al., 2007*). Nanobody sequences were then amplified by PCR in two steps for the generation of a yeast display library. PCR products were ethanol precipitated, and 160 µg of resulting PCR product and 40 µg linearized pCTCON2 vector (*Colby et al., 2004*) were co-transformed into electrocompetent EBY100 yeast (Invitrogen). Following transformation, cells were recovered in 1 L SDCAA at 30°C. Ten-fold serial dilutions were plated onto SDCAA, grown for 3 days at 30°C, and colonies were counted to determine library size. The resulting library was determined to contain 4.8 × 10$^6$ transformants. The library was grown at 30°C until it reached ten-fold increase over the initial calculated diversity, then pelleted at 3000 x g, and stored in 10% glycerol at −80°C.

## Nanobody selection by yeast surface display

For the first round of selection, 5 × 10$^7$ yeast displaying the nanobody library were cleared of non-specific binders to Alexa-647 conjugated anti-protein C antibody as described above for the yeast-displayed chemokine library. These cleared yeast were then washed with staining buffer and stained with 1 µM US28Nb7 for 2 hr at 4°C. Magnetic selection then proceeded as described above for the yeast displayed chemokine library.

For the second round of selection, two-color FACS was performed. 1 × 10$^7$ induced yeast were washed with staining buffer and stained with 200 nM US28Nb7ΔN for 2 hr at 4°C. The yeast were then washed with staining buffer and stained with Alexa-647 conjugated anti-protein C and Alexa-488 conjugated anti-Myc antibodies (Cell Signaling) for 15 min at 4°C. Yeast were washed again with staining buffer and the Alexa-647 and −488 double-positive cells were purified using a FACS Jazz cell sorter (BD Biosciences). Post-sorted yeast were resuspended in SDCAA medium and cultured at 30°C.

The third round of selection also employed two-color FACS. 1 × 10$^7$ induced yeast were washed with staining buffer and stained with 10 µM US28Nb11 for 2 hr at 4°C. The yeast were then washed with staining buffer and stained with Alexa-647-conjugated anti-protein C and Alexa-488-conjugated anti-Myc antibodies for 15 min at 4°C. Yeast were washed again and both double-positive cells and Alexa-488 singly positive cells were purified separately to distinguish nanobodies that bind both

US28Nb7 and US28Nb11 from those that are selective. Nanobody cDNA was prepared from each of the post Round three library samples, transformed into *E. coli* and sequenced, yielding nanobody B1.

## Nanobody purification

Nanobody B1 was cloned into pAcGP67A insect expression vector (BD Biosciences) with a C-terminal 6xHis tag. Baculovirus was added to High Five cells at a density of $2 \times 10^6$ cell ml$^{-1}$ and incubated for 60 hr at 28°C. Collected culture media was conditioned with 50 mM Tris-HCl pH 8.0, 1 mM NiCl$_2$, 5 mM CaCl$_2$ and the subsequent precipitate was cleared via centrifugation. The media was then incubated with Ni-NTA resin (QIAGEN) at room temperature for 3 hr and eluted in HBS with 200 mM imidazole. The elution was further purified by size exclusion chromatography before it was concentrated and aliquoted for future protein complex formation.

## CX3CL1.35-US28Nb7-Nb B1 complex purification

CX3CL1.35 was expressed as described for chemokines above. After the initial centrifugation, the HEK293S GnTI- culture supernatant was filtered and stirred at 4°C overnight with 10 ml protein A agarose (Sigma). The protein A agarose was then collected by filtration and washed with HBS.

US28Nb7 was expressed and solubilized as described above. Following centrifugation, protein A-immobilized CX3CL1.35–3C-Fc was added to the solubilized lysate and mixed by rotating overnight at 4°C. The protein A resin was collected by filtration, washed with HBS and incubated at room temperature for 90 min with ~400 ng 3C protease per liter of initial culture to release the CX3CL1.35-US28Nb7 complex. Anti-FLAG M1 affinity resin was used to further purify the complex and remove excess CX3CL1.35. The complex was eluted from the anti-FLAG M1 affinity resin with 0.2 mg ml$^{-1}$ FLAG peptide and 5 mM EDTA.

Nanobody B1 was added to the CX3CL1.35-US28Nb7 complex at a molar ratio of 1.25:1. The ternary complex was isolated and desalted by size exclusion chromatography using a buffer containing HBS, 0.02% (w/v) DDM, and 0.004% (w/v) CHS. Complex formation was confirmed by SDS-PAGE. The complex was then concentrated to 28 mg ml$^{-1}$, aliquoted, and flash-frozen before crystallization trials.

## US28Nb7 purification

US28Nb7 purification for crystallization proceeded as described above for staining experiments. The protein was concentrated to 28 mg ml$^{-1}$, aliquoted, and flash-frozen before crystallization trials.

## Crystallization, data collection and structure determination

Crystallization was performed by *in meso* method with the Gryphon LCP robot (Art Robbins Instruments, ARI, Sunnyvale CA). In the case of CX3CL1.35-US28Nb7-Nb B1, 1:150 (w/w) carboxypeptidases A and B (Sigma-Aldrich) were added to each protein aliquot and the mixture was incubated for 30 min at room temperature to truncate disordered C-terminal residues in situ.

US28Nb7 and CX3CL1.35-US28Nb7-Nb B1 were then reconstituted into LCP by mixing with prepared and pre-warm 10:1 (w/w) monoolein:cholesterol (Sigma-Aldrich) mixture in a 1:1.5 (w/w) protein:lipid ratio. All samples were mixed with the coupled ARI syringes at least 100 times.

The resulting LCP was dispensed in 20–50 nL onto a glass plate with a 96-well double-sided spacer tapes. The drops were then overlaid with 650 μL of crystallization buffers summarized in *Figure 5—source data 1* and sealed with a cover glass. Crystals grown at 16°C or 20°C in 4–7 days were harvested using MicroMesh loops (MiTeGen, Ithaca NY) together with the LCP drop, flash frozen and stored in liquid nitrogen. X-ray data collection was performed at Advanced Photon Source GM/CA beamline 23ID-D. The datasets were initially processed with XDS (*Kabsch, 2010*). The phases were determined by molecular replacement using Phaser (*McCoy et al., 2007*) with the US28-Nb7 complex and Nb7 with truncation of complementarity determining regions (both are from PDB: 4xt1) as search models. The structures were further built and fixed using Coot (*Emsley et al., 2010*) and refined with Phenix (*Adams et al., 2010*; *Afonine et al., 2012*; *Echols et al., 2012*) using individual atomic displacement parameters. In the early stages of refinement, reference torsion restraints were generated (*Headd et al., 2012*) from the CX3CL1/US28/Nb7 crystal structure (PDB: 4xt1). Data collection and refinement statistics are also summarized in *Figure 5—source data 1*.

Atomic contacts were analyzed and structural figures were prepared using PyMol (*Schrodinger, 2010*) and structural alignments of the US28 transmembrane helices and other atomic coordinate transformations were performed using LSQMAN. Software used in this project was installed and configured by SBGrid (*Morin et al., 2013*).

## Acknowledgements

We thank Jamie B Spangler and Juan L Mendoza for helpful discussion and technical assistance. We also acknowledge the beamline resources and staff of Advanced Photon Source GM/CA beamlines 23-ID-B and 23-ID-D and Stanford Synchrotron Radiation Lightsource beamline 12–2. We acknowledge support from the American Heart Association (TFM), Danish Research Council (KS and MMR), Human Frontier Science Program (NT), Cancer Research Institute (JSB), Claudia Adams Barr Program for Innovative Cancer Research (JRI), Howard Hughes Medical Institute (KCG), NIH R01 AI125320 (KCG), NovoNordisk Foundation (MMR), Hoslev Foundation (MMR), Aase and Einer Danielsen Foundation (MMR), Lundbeck Foundation (MMR), Gangsted Foundation (MMR), Director's Pioneer Award (HLP), and Lustgarten Foundation (HLP). Structure factors and coordinates have been deposited in the Protein Data Bank with accession codes 5wb1 and 5wb2.

## Additional information

### Funding

| Funder | Grant reference number | Author |
| --- | --- | --- |
| National Institutes of Health | R01 AI125320 | K Christopher Garcia |
| Howard Hughes Medical Institute | | K Christopher Garcia |
| American Heart Association | Postdoctoral Fellowship | Timothy F Miles |
| Uddannelses- og Forsknings-ministeriet | | Katja Spiess<br>Mette M Rosenkilde |
| Human Frontier Science Program | Postdoctoral Fellowship | Naotaka Tsutsumi |
| Novo Nordisk | | Mette M Rosenkilde |
| Hoslev Foundation | | Mette M Rosenkilde |
| Aase and Einer Danielsen Foundation | | Mette M Rosenkilde |
| Lundbeckfonden | | Mette M Rosenkilde |
| Gangstedfonden | | Mette M Rosenkilde |
| National Institutes of Health | Director's Pioneer Award | Hidde L Ploegh |
| Lustgarten Foundation | | Hidde L Ploegh |
| Cancer Research Institute | Irvington Postdoctoral Fellowship | John S Burg |
| Claudia Adams Barr Program for Innovative Cancer Research | | Jessica R Ingram |

The funders had no role in study design, data collection and interpretation, or the decision to submit the work for publication.

### Author contributions

Timothy F Miles, Conceptualization, Formal analysis, Investigation, Visualization, Methodology, Writing—original draft, Writing—review and editing; Katja Spiess, Gertrud M Hjorto, Formal analysis, Investigation, Writing—review and editing; Kevin M Jude, Naotaka Tsutsumi, John S Burg, Jessica R Ingram, Deepa Waghray, Olav Larsen, Investigation, Writing—review and editing; Hidde L Ploegh, Mette M Rosenkilde, Supervision, Funding acquisition, Writing—review and editing; K Christopher

Garcia, Conceptualization, Supervision, Funding acquisition, Project administration, Writing—review and editing

## Author ORCIDs
Timothy F Miles https://orcid.org/0000-0001-6591-3271
Kevin M Jude https://orcid.org/0000-0002-3675-5136
Naotaka Tsutsumi http://orcid.org/0000-0002-3617-7145
Olav Larsen https://orcid.org/0000-0001-9054-4690
K Christopher Garcia https://orcid.org/0000-0001-9273-0278

## Decision letter and Author response
Decision letter https://doi.org/10.7554/eLife.35850.040
Author response https://doi.org/10.7554/eLife.35850.041

# Additional files

## Supplementary files
• Transparent reporting form
DOI: https://doi.org/10.7554/eLife.35850.028

## Data availability
Sequencing data have been deposited in SRA at NCBI under accession codes: SAMN08581530, SAMN08581531, and SAMN08581532. Diffraction data have been deposited in PBD under the accession codes 5wb1 and 5wb2.

The following datasets were generated:

| Author(s) | Year | Dataset title | Dataset URL | Database, license, and accessibility information |
|---|---|---|---|---|
| Miles TF | 2018 | Naive Unselected Library | https://www.ncbi.nlm.nih.gov/biosample/8581530 | Publicly available at the NCBI Biosample database (Acc no. 8581530) |
| Miles TF | 2018 | US28Nb7 Round2 | https://www.ncbi.nlm.nih.gov/biosample/8581531 | Publicly available at the NCBI Biosample database (Acc no. 8581531) |
| Miles TF | 2018 | US28Nb11 Round3 | https://www.ncbi.nlm.nih.gov/biosample/8581532 | Publicly available at the NCBI Biosample database (Acc no. 8581532) |
| Miles TF | 2018 | Diffraction data | http://www.rcsb.org/pdb/search/structid-Search.do?structureId=5wb1 | Publicly available at the RCSB Protein Data Bank (accession no: 5wb1) |
| Miles TF | 2018 | Diffraction data | http://www.rcsb.org/pdb/search/structid-Search.do?structureId=5wb2 | Publicly available at the RCSB Protein Data Bank (accession no: 5wb2) |

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
