## [Decision Letter]

Thank you for submitting your article "Viral GPCR US28 can signal in response to chemokine agonists of nearly unlimited structural degeneracy" for consideration by *eLife*. Your article has been reviewed favorably by three peer reviewers, and the evaluation has been overseen by John Kuriyan (Senior Editor and also Reviewing Editor). The reviewers have opted to remain anonymous.

The reviewers have discussed the reviews with one another and the Reviewing Editor has drafted this decision to help you prepare a revised submission.

Review:

In this manuscript, the authors explored the interactions of chemokine CX3CL1 and the viral GPCR US28, particularly focused on the N-terminal seven amino acid residues of CX3CL1. This N-terminal segment binds within the receptor transmembrane pocket. The authors explored the influence of the bound N-terminal segment on receptor activation.

The authors utilized US28 fused with nanobodies that stabilize apo-US28 in active-like conformations, and screened a yeast-displayed library of CX3CL1 in which the N-terminal seven residues are randomized. A large number of chemokine variants that bind to the receptor and signal are identified. These variants have essentially no sequence similarity in the N-terminal segment of CX3CL1, and signal to varying extents.

A structure of a CX3CL variant complexed to US28 provides a framework to understand the interactions that various sequences can form with different residues in the transmembrane pocket. The structure reveals modest conformational changes in the receptor compared to a previous complex structure and the apo receptor structure, but the chemokine amino terminus itself adopts a highly distinct conformation. Two chimeric chemokines between the globular domain of CX3CL1 and the N-terminus of vMIP-II (NVF) and CCR5 (N5F) have weak Ca^2+^ and migration activity. What is novel is the conversion of vMIP-II antagonist to an agonist in the NVF chimera.

By integrating all these data, the authors conclude that the specific molecular contacts of the amino terminus are largely dispensable for receptor activation in the case of US28. Overall, this is a rigorously-performed and well-written study that is likely to be of considerable interest to the field, particularly those interested in protein ligand-GPCR interactions controlling receptor signaling.

Essential points to address:

1) The design of the library allows broad, but non-uniform, sampling of possible amino acids at each position. The analysis of resulting frequencies is difficult to interpret, since the input frequencies of amino acids are not equal. For example, the authors point to the high frequency of hydrophobic residues at position 1 as evidence that signal peptidase cleavage efficiency is not a major factor in sequence enrichment. However, the input frequency expected here is skewed toward hydrophobic residues by library design, with F, I, L, and V collectively expected to occur about four times as often as either D or E. There may be additional bias if mixed bases are not incorporated with equal efficiency, or if library amplification PCR skewed sequence distributions. As a result, comparison of input and output frequencies is necessary to draw conclusions about the relative enrichment for particular features. It would have been straightforward to have sequenced the input libraries as well, and this would allow a direct assessment of what the receptor binding and FACS is really selecting. Are these data available? This would substantially improve the analysis presented. If not, could some calibration experiments be done?

2) The reviewers feel that single-point signaling assays, like those presented in Figure 3, offer limited information, and a lack of activity may reflect either antagonism or insufficient potency. To properly characterize these new binders the authors should at least perform dose-response signaling analysis, ideally alongside a binding assay to measure the affinities of these clones. To show biased signaling, there should be dose-response curves of two assays – for G protein-coupling pathway (US28-induced IP3 turnover or Ca^2+^ response) and β-arrestin pathway (β-arrestin-recruitment or US28-induced migration assay). If these assays cannot be done, then the claims about biased signaling should be removed.

3) It would be helpful to show the affinities to US28 of different ligands, such as ligands typical of those selected. Indeed, the potency of these variants is not shown in the present manuscript.

4) In general, binding to the receptor N-terminus (mostly by the globular domain of the cytokine) plays a role in the affinity of the receptor-chemokine interaction. The conversion of antagonist vMIP-II to agonist by exchanging the globular domain with CX3CL1 suggests signaling roles of the globular domain interactions. To what extent is binding determined by the chemokine's amino-terminal peptide vs. the folded C-terminal domain? The authors state that a poly-Gly N-terminus is not sufficient (Results and Discussion, fifth paragraph), which helps argue that the peptide is a key determinant, but they don't show data. Given the fact that the whole paper is about the role of the amino terminal peptide in binding and signaling, it is important to actually show data for this, and the statement "data not shown" in the submitted version of the manuscript implies that they've already done the experiment.

Less critical points:

1) The authors provide an excellent description of the differences between variant 1.35 and the wt structure. The difference between 1.35 and wt interactions with the receptor are profound (Figure 4—figure supplement 2), especially in the globular section of the chemokine. Are there crystal contacts of the 1.35 variant or of the previously published wt CX3CL1 complex that could influence the globular portion with respect to the receptor?

2) The authors conclude that "steric bulk of the ligand is more important than specific bonding chemistries" in the TM cavity. They also should discuss the steric bulk in the context of GPCR dynamics, which is known to also be important in activation and signaling. Although the crystallography is impressive, the authors should be careful of over-interpreting their results as the fusion with Nb will prevent conformational changes to occur and signaling proteins to bind. Unfortunately, although the authors offer a "caveat" about their structures, the over-interpretation occurs.

3) There is an invariant Pro at position 3 for all of the sequences in Figure 3C and D. It is difficult to see anything about the structural features of a proline vis-à-vis the interaction with the pocket in the figures.

4) Please explain how "strains on the walls of the binding pocket" results in activation, if the mechanism is similar to Kobilka's mechanism for adrenergic activation.

5) Should the amino acid possibilities in position 1 shown in Figure 2—figure supplement 2 include also lysine (codon AAA)?

---

## [Author Response]

Essential points to address:

1) The design of the library allows broad, but non-uniform, sampling of possible amino acids at each position. The analysis of resulting frequencies is difficult to interpret, since the input frequencies of amino acids are not equal. For example, the authors point to the high frequency of hydrophobic residues at position 1 as evidence that signal peptidase cleavage efficiency is not a major factor in sequence enrichment. However, the input frequency expected here is skewed toward hydrophobic residues by library design, with F, I, L, and V collectively expected to occur about four times as often as either D or E. There may be additional bias if mixed bases are not incorporated with equal efficiency, or if library amplification PCR skewed sequence distributions. As a result, comparison of input and output frequencies is necessary to draw conclusions about the relative enrichment for particular features. It would have been straightforward to have sequenced the input libraries as well, and this would allow a direct assessment of what the receptor binding and FACS is really selecting. Are these data available? This would substantially improve the analysis presented. If not, could some calibration experiments be done?

The reviewers are correct to note that the library allows a non-uniform distribution and that it is possible for skew to arise in its construction. The naïve library was deep sequenced after construction and prior to selection. The resulting amino acid distribution by position was used as the baseline against which to measure enrichment or depletion during selections. This data is presented in Figure 2—figure supplement 3. The legend for this figure has been updated and expanded to clarify this point. All conclusions in the text, such as those raised by the reviewers above, are supported by this analysis.

2) The reviewers feel that single-point signaling assays, like those presented in Figure 3, offer limited information, and a lack of activity may reflect either antagonism or insufficient potency. To properly characterize these new binders the authors should at least perform dose-response signaling analysis, ideally alongside a binding assay to measure the affinities of these clones. To show biased signaling, there should be dose-response curves of two assays – for G protein-coupling pathway (US28-induced IP3 turnover or Ca^2+^ response) and β-arrestin pathway (β-arrestin-recruitment or US28-induced migration assay). If these assays cannot be done, then the claims about biased signaling should be removed.

The reviewers raise an important point that we clarify in the revision. Ca^2+^ response and US28-induced migration dose-response analyses been included in Figure 1—figure supplements 1 and 2 and Figure 3—figure supplements 1 and 2. There were several confounding factors that hampered our ability to achieve saturation of the dose-response curves. First, negative effects on cell viability preclude data inclusion at concentrations of 1 μM chemokine and greater. Second, the rapid constitutive internalization rates of US28 make it very hard to fully complete the dose-response curves at the highest concentrations. While many of these curves do not reach saturation and allow for precise quantification of bias factors, the data are strongly suggestive of bias in US28’s signaling and we are careful not to make quantitative claims about bias, and only limit our conclusions about bias to qualitative differences. We have indicated the need for caution with regard to these data in the main text (subsection “Chemokine-induced US28 signaling”, second paragraph) but believe the data justify qualitative discussion. We have also inserted the sentence “Because the high basal signaling of US28 results in a narrow dynamic range in which to measure differences between agonists, we can only make qualitative conclusions about signaling bias.”

3) It would be helpful to show the affinities to US28 of different ligands, such as ligands typical of those selected. Indeed, the potency of these variants is not shown in the present manuscript.

Affinities of selected CX3CL1 library variants are now shown in Figure 3—figure supplement 3 and the main text has been modified to address this (subsection “The sequence space of chemokine agonism”, last paragraph). Clones selected from the CX3CL1 library bind equivalently to wild type CX3CL1 when yeast displayed and stained with nanobody-stabilized receptor. Radioligand competition experiments of the CX3CL1 library variants reveal weaker competition with wild type CX3CL1 (i.e. a lower affinity in heterologous competition binding experiments) than the wild type chemokine competes with itself (i.e. in homologous competition binding experiments). This apparent low affinity in the heterologous competition binding setting is precedented by wild type CCL3 and CCL5 (Figure 1A) that are unable to displace radiolabeled CX3CL1. Importantly this impaired competition does not reflect low affinity for the unliganded receptor (as described earlier for US28 in Kledal, Rosenkilde, and Schwartz, 1998 and for the NK1 receptor (Rosenkilde et al., 1994, JBC).

4) In general, binding to the receptor N-terminus (mostly by the globular domain of the cytokine) plays a role in the affinity of the receptor-chemokine interaction. The conversion of antagonist vMIP-II to agonist by exchanging the globular domain with CX3CL1 suggests signaling roles of the globular domain interactions.

Additional language has been added to the main text to address this possibility (subsection “Chemokine-induced US28 signaling”, last paragraph). It is possible that the globular domains contribute to signaling only insofar as they align how the chemokine amino terminus is positioned within the receptor binding site. A difference in chemokine N-terminus positioning is likely given the CC to CX3C conversion in NVF.

To what extent is binding determined by the chemokine's amino-terminal peptide vs. the folded C-terminal domain?

The balance of contributions from sites 1 and 2 to chemokine affinity for the nanobody-stabilized receptor are chemokine-specific. When the receptor N-terminus is removed, CX3CL1 affinity for the receptor is largely unchanged. CCL3 on the other hand loses all binding to US28Nb7. We believe that nanobody7 is enhancing US28’s affinity for the amino termini of certain chemokines. We have included a new figure supplement (Figure 2—figure supplement 2) and language in the main text addressing this point (subsection “The sequence space of chemokine agonism”, second paragraph).

The authors state that a poly-Gly N-terminus is not sufficient (Results and Discussion, fifth paragraph), which helps argue that the peptide is a key determinant, but they don't show data. Given the fact that the whole paper is about the role of the amino terminal peptide in binding and signaling, it is important to actually show data for this, and the statement "data not shown" in the submitted version of the manuscript implies that they've already done the experiment.

This data is now included as Figure 2—figure supplement 4. The combination of the loss of CX3CL1 affinity when the chemokine amino terminus is changed with the lack of effect when the receptor N-terminus is completely removed convince us that CX3CL1 affinity for US28Nb7 is largely derived from site 2 interactions.

Less critical points:

1) The authors provide an excellent description of the differences between variant 1.35 and the wt structure. The difference between 1.35 and wt interactions with the receptor are profound (Figure 4—figure supplement 2), especially in the globular section of the chemokine. Are there crystal contacts of the 1.35 variant or of the previously published wt CX3CL1 complex that could influence the globular portion with respect to the receptor?

The reviewers are correct to note that lattice contacts may influence the conformation of the chemokine. We have added data (Figure 4—figure supplement 4) and discussion (subsection “Structural basis for US28’s chemokine promiscuity”) of this possibility. This includes arguments that the CX3CL1.35 complex most likely has a unique orientation that is captured, rather than induced, by the crystal lattice.

2) The authors conclude that "steric bulk of the ligand is more important than specific bonding chemistries" in the TM cavity. They also should discuss the steric bulk in the context of GPCR dynamics, which is known to also be important in activation and signaling. Although the crystallography is impressive, the authors should be careful of over-interpreting their results as the fusion with Nb will prevent conformational changes to occur and signaling proteins to bind. Unfortunately, although the authors offer a "caveat" about their structures, the over-interpretation occurs.

It is difficult to see where the reviewers think we are over-interpreting. At the same time, none of our methods are sensitive to issues of dynamics.

3) There is an invariant Pro at position 3 for all of the sequences in Figure 3C and D. It is difficult to see anything about the structural features of a proline vis-à-vis the interaction with the pocket in the figures.

This proline’s conservation is merely a function of its having been used as a hallmark by which to delineate this family of related sequences. The structure itself does not point to a special role for this residue.

4) Please explain how "strains on the walls of the binding pocket" results in activation, if the mechanism is similar to Kobilka's mechanism for adrenergic activation.

There is no unifying mechanism for GPCR activation at the site of ligand binding; Activation pathways across families converge only near the G protein coupling region (doi: 10.1038/nature19107). In a system sampling a broad conformational landscape with shallow energetic barriers to interconversion between states, steric obstruction in the binding site may bias the distribution of states sampled by the GPCR. The main text highlights features common to chemokine receptors that would support the adoption of such a mechanism (subsection “Conclusions”).

5) Should the amino acid possibilities in position 1 shown in Figure 2—figure supplement 2 include also lysine (codon AAA)?

Yes, this has been corrected in the new Figure 2—figure supplement 3.